# TREE-STRUCTURE SEGMENTATION FOR LOGISTIC REGRESSION

## ABSTRACT

The decision for a financial institution to accept or deny a loan is based on the probability of a client paying back their debt in time. This probability is given by a model such as a logistic regression, and estimated based on, *e.g.*, the clients' characteristics, their credit history, the repayment performance. Historically, different models have been developed on different markets and/or credit products and/or addressed population. We show that this amounts to modelling default as a mixture model composed of a decision tree and logistic regression on its leaves (thereafter "logistic regression tree"). We seek to optimise this practice by considering the population to which a client belongs as a latent variable, which we will estimate. After exposing the context, the notations and the problem formalisation, we will conduct estimation using a Stochastic-Expectation-Maximisation (SEM) algorithm. We will finally show the performance on simulated data, and on real retail credit data from [COMPANY], as well as real open-source data.

## 1 INTRODUCTION AND NOTATIONS

### 1.1 CONTEXT

[COMPANY], like most financial institutions, has a relatively automatic procedure to accept or deny loans and estimate its capital requirements. The procedure is based on credit scores. A client fills in a questionnaire with socio-demographic information and banking behavioural questions, which answers are used to compute a score. This score determines the financing of the client and the necessary impairment for the bank to be ready in case of a potential default. The score is learned on past clients' characteristics (from the questionnaire), which we denote by $\boldsymbol{x}$, and the repayment in time, or not, of their loan which we denote by $y \in \{0, 1\}$ (where 1 represents the default). The score is directly proportional to the probability $p(1|\boldsymbol{x})$ of the client not paying back the loan in time, which is estimated with a model $\{p_{\boldsymbol{\theta}}(y|\boldsymbol{x})\}_{\boldsymbol{\theta} \in \Theta}$. A parametric family $\Theta$ is chosen (usually logistic regression) and the optimal parameter $\hat{\boldsymbol{\theta}}$ in this family ($\hat{\boldsymbol{\theta}} \in \Theta$) is estimated from an $n$-sample $(\mathbf{x}, \mathbf{y}) = (\boldsymbol{x}_i, y_i)_1^n$, usually using a maximum likelihood approach. Such a model is relatively weak, in the sense that the hypothesis space is too restricted to fit the whole clientele of big financial institutions.

### 1.2 A MODEL FOR EACH SEGMENT OF CLIENTS

Most financial institutions address multiple markets, *e.g.* automobile, home appliances, or partners who sell such products, and different populations of clients (professionals, organisations, agriculture, private clients). We call "segment" such a sub-population, and denote it by $c \in \mathcal{C} = \{1, \ldots, K\}$, where $K$ denotes the total number of segments.

Formally, we have a vector of customer characteristics $\boldsymbol{X} = (X^j)_1^d$, made of $d$ features, either continuous (*i.e.* valued in $\mathbb{R}$) or categorical (*i.e.* valued in $\{1, \ldots, m_j\}$ without order). The aim is to predict the default $Y \in \{0, 1\}$ from an observation $\boldsymbol{x}$. These features differ depending on the segment of the population, for instance "time since creation of the company" is a feature which does not apply to private clients. However, for simplicity, in the rest of the paper, we will assume that all $d$ features are shared by all segments. This leads to little loss in generality since continuous features can be discretized and a "Not Relevant" level can be introduced for categorical features; additionally we can resort to feature selection at the segment level (see Section 4.6).

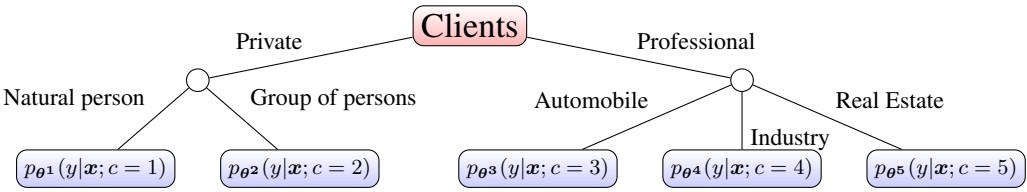

Figure 1: Example of model segmentation.

Subsequently, financial institutions create different predictive models $\{p_{\boldsymbol{\theta}^c}(y|\boldsymbol{x})\}_1^K$ for each population $c$, where $\boldsymbol{\theta}^c$ denotes the coefficient used for segment $c$ (with potential null entries), as shown in Figure 1, which leads to $K$ models. This means we learn "expert" logistic regression models on separate "segments" of clients arranged in a tree.

Since this structure is inherited from past a priori decisions, it is likely to be sub-optimal; hence we seek to optimise the performance on the whole population. To this end, we formalise the data generating process in the next section.

### 1.3 FORMALISATION OF THE DATA GENERATING PROCESS

We assume that the model in Figure 1 used by financial institutions accurately depicts the data generation, *i.e.* for a given client $\boldsymbol{x}$, there exists a segment $c$ and a logistic regression parameter $\boldsymbol{\theta}^c$ for which the default $y$ is drawn from $p_{\boldsymbol{\theta}^c}(\cdot|\boldsymbol{x};c)$. In other words, we assume that this model is well-specified. We denote by $C \sim p(\cdot)$ the random variable valued in $\{1, \ldots, K\}$ which corresponds to the assignment to a group (the tree's leaves in Figure 1). $C$ specifies both the distribution of the predicting variables, *i.e.* $\boldsymbol{x}|c \sim p(\cdot|c)$ and the default law for each group, which we suppose to be logistic, *i.e.* $Y|\boldsymbol{x}, c \sim p_{\boldsymbol{\theta}^c}(\cdot|\boldsymbol{x}, c)$.

Just like for Gaussian mixture models, we seek to estimate $p(c|\boldsymbol{x})$ (a simple proportion in the latter example), which will subsequently allow estimation of $p_{\boldsymbol{\theta}^c}(y|\boldsymbol{x};c)$. Current procedures are described in the next section.

### 1.4 AD HOC "TWO-STAGES" PRACTICE

The *ad-hoc* methods rely on "two-stages" procedures: first optimising the segmentation, then learning the separate logistic regression on each segment. The segmentation is done by practitioners using simple unsupervised "clustering" techniques such as Principal Component Analysis (PCA) and its refinements. In presence of (possibly only) categorical features, the Multiple Correspondence Analysis (MCA), by Lebart et al. (1995) or the Factor Analysis of Mixed Data (FAMD), by Pagès (2014) can be more appropriate. Practitioners then visually assess whether clusters appear on the projection of samples onto the 2-3 first Principal Components like in the examples of Appendix A, thus resulting in a qualitative, clustering-like technique, which often performs poorly.

In Section 2 we review the existing approaches to create logistic regression trees. In Section 3 we formalise the problem of determining the best logistic regression tree as a mixture model and propose an estimation strategy in Section 4. We devote Section 5 to numerical experiments on simulated data, and Section 6 to experiments on real data.

## 2 LITERATURE REVIEW OF EXISTING DIRECT APPROACHES: LOGISTIC REGRESSION TREES

The first research work focusing on a similar problem than the present one seems to be LOTUS, by Chan & Loh (2004), where logistic regression trees are constructed so as to select features to split the data on the tree's nodes which break the linearity assumption of logistic regression. Its authors' motivation is that logistic regression has a fixed parameter space, defined by the number of input features, whereas trees adapt their flexibility (*i.e.* depth) to the sample size $n$. Thus, they search for trees which leaves are logistic regressions with a few continuous features and which intermediate nodes (found *via* an appropriate $\chi^2$ test) split the population based on categorical or

continuous features which relationship to the log-odd ratio of $y$ is not linear (*i.e.* features that would perform poorly in a logistic regression). Their optimised criterion is the sum of the log-likelihoods of the logistic regression on the tree's leaves. This leads to overfitting which requires the tree to be pruned (as is classical for decision trees) using a method closely related to the one developed in the classical CART algorithm by Breiman et al. (1984).

The second approach closely related to our industrial problem is named LMT, by Landwehr et al. (2005). Its authors' approach differs however from LOTUS in that they rely on a boosting approach derived from the LogitBoost algorithm by Friedman et al. (2000) to estimate the logistic regression, and an adaptation of the classical C4.5 algorithm by Quinlan (2014) to grow the tree. The two central ideas behind their usage of the LogitBoost algorithm are that: it allows (1) to perform feature selection *via* a stage-wise-like process where one feature enters the model at each step, and (2) to recursively "refine" the logistic regression by boosting the logistic regression fitted at a node's parent. Indeed, a first logistic regression is fitted at the tree's root via LogitBoost using all observations $(\mathbf{x}, \mathbf{y})$, which is further boosted separately at its subsequent children nodes on sub-populations, say $((\mathbf{x}^1, \mathbf{y}^1), (\mathbf{x}^2, \mathbf{y}^2))$ and so on. This most probably induces less parameter estimation variance in each leaf since they partly benefit from samples not in their leaf but used to fit the parents' logistic regression, and it is fast. The resulting tree must also be pruned and either a tactic similar to the classical tree algorithm CART, or cross-validation, or the AIC criterion (in a refinement of the method proposed by Sumner et al. (2005)) are used.

Lastly, a third approach is MOB, by Zeileis et al. (2008). Their algorithm consists in fitting the chosen model (in our case, logistic regression) for all observations at the current node and decide to split these into subsets based on a correlation measure (several such measures are proposed) of the residuals of the current model $(cor(\mathbf{x}_j^c, y - p_{\hat{\boldsymbol{\theta}}^c}(\mathbf{y}^c|\mathbf{x}^c)))$. The procedure is repeated until no significant "correlation" is detected. Similarly to LOTUS and contrary to C4.5, MOB performs, for binary splits and when confronted to a categorical feature $j$ having $m_j$ levels, $2^{m_j}$ tests. Finally, the number of segments per split is searched exhaustively. Thus, it is computation intensive.

To sum up, these direct approaches produce the sought tree-structure of Figure 1 with different algorithms: LOTUS only considers continuous features in the leaves and relies on a $\chi^2$ test to select the splits. LMT relies on C4.5 and boosting to grow the tree and estimate the logistic regression respectively. MOB estimates a logistic regression at each node and chooses splits according to a correlation to its residuals. In addition to MOB and LMT, there is a vast literature on Hierarchical Mixtures of Experts from Jordan & Jacobs (1994) which have a "tree-like" structure but for which all "experts" (our logistic regressions) output predictions which are thereafter weighted by the tree structure. Despite "looking" similar, the approaches differ because our desired segmentation means we want to rely on a single expert for each instance, and we wish that this expert is chosen by a classical decision tree (in particular, univariate splits at each node). We formalise now the problem as a model selection problem.

## 3 Logistic regression trees: a difficult optimisation problem

In the *ad-hoc* method, the segments $(c_i)_1^K$ were determined *a priori* using historical or practical reasons as shown in Figure 1. As we aim at optimising the segmentation, it is desirable to find the probability of belonging to each segment $c$, and to fit the model $p_{\boldsymbol{\theta}^c}(y|\boldsymbol{x}, c)$ on each segment. The total number of segments $K$ is also to be determined. This amounts to a mixture model:

$$p(y|\boldsymbol{x}) = \sum_{c=1}^{K} p_{\boldsymbol{\theta}^c}(y|\boldsymbol{x}, c) p(c|\boldsymbol{x}). \tag{1}$$

The approach we take considers the real segment $C^\star$ as a latent random feature. Each observation belongs to one segment only, thus $p(c|\boldsymbol{x})$ is non-zero only for $c^\star$. Subsequently, denoting by $\mathbf{x}^{c^\star}$ the subset of observations for which $c = c^\star$ and by $c_i^\star$ the segment of an observation $\boldsymbol{x}_i$ we have:

$$p(\mathbf{x}, \mathbf{y}) = \sum_{c=1}^{K^\star} p(\mathbf{y}|\mathbf{x}; c)p(c|\mathbf{x})p(\mathbf{x}) = \prod_{i=1}^{n} \sum_{c=1}^{K^\star} p(y_i|\boldsymbol{x}_i; c) \underbrace{p(c|\boldsymbol{x}_i)}_{=0 \text{ for } c \neq c^\star} p(\boldsymbol{x}_i)$$

$$= \prod_{i=1}^{n} p(y_i|\boldsymbol{x}_i; c_i^\star)p(\boldsymbol{x}_i) = \prod_{c^\star=1}^{K^\star} p(\mathbf{y}^{c^\star}|\mathbf{x}^{c^\star}; c^\star)p(\mathbf{x})$$

$$= \prod_{c^\star=1}^{K^\star} \int_{\Theta^{\star,c^\star}} p_{\boldsymbol{\theta}^{\star,c^\star}}(\mathbf{y}^{c^\star}|\mathbf{x}^{c^\star})p(\boldsymbol{\theta}^{\star,c^\star}|c^\star)d\boldsymbol{\theta}^{\star,c^\star}p(\mathbf{x}).$$

$$\ln p(\mathbf{x}, \mathbf{y}) = \sum_{c^\star=1}^{K^\star} \int_{\Theta^{\star,c^\star}} \ln p_{\boldsymbol{\theta}^{\star,c^\star}}(\mathbf{y}^{c^\star}|\mathbf{x}^{c^\star})p(\boldsymbol{\theta}^{\star,c^\star}|c^\star)d\boldsymbol{\theta}^{\star,c^\star} + \ln p(\mathbf{x}) \qquad (2)$$

$$\approx -\sum_{c^\star=1}^{K^\star} \text{BIC}(\boldsymbol{\theta}^{\star,c^\star})/2 + O(K^\star) + \ln p(\mathbf{x}),$$

where BIC stands for the Bayesian Information Criterion (similar to AIC, see Neath & Cavanaugh (2012)). Since in our application, the number of sample $n \approx 10^5$ is large and the number of desired segments $K^\star \approx 10$ is low, we use the following criterion to select a segmentation:

$$(\hat{K}, \hat{c}) = \arg\min_{K,c} \sum_{c=1}^{K} \text{BIC}(\hat{\boldsymbol{\theta}}^c). \qquad (3)$$

The difficulty in optimising Equation 3 directly lies in the discrete nature of $c$ given $\boldsymbol{x}$. This highly-combinatorial discrete problem is relaxed by approximating door functions $p(c|\boldsymbol{x})$ with a "smooth" proxy $p_{\boldsymbol{\beta}}(c|\boldsymbol{x})$ and relying on Markov Chain Monte Carlo (MCMC) methods. As we will rely on decision trees, the $\boldsymbol{\beta}$ "parameter" will stand for the tree's split features and cut points.

## 4 ESTIMATING LOGISTIC REGRESSION TREES

As Equation 1 is a mixture model with a latent variable, it seems natural to resort to an Expectation-Maximization (EM) algorithm, which will be exposed in the next section. We will then mitigate two downsides of this approach by relying on a Stochastic-Expectation-Maximization (SEM) algorithm in the subsequent section, and finally discuss how we obtain a logistic regression tree from all the candidates that these two algorithms provide.

### 4.1 A CLASSICAL EM ESTIMATION STRATEGY

We would like to maximise the following likelihood, derived from Equation 1, both in terms of the segmentation and the resulting logistic regressions:

$$\ell(\boldsymbol{\beta}, (\boldsymbol{\theta}^c)_1^K; \mathbf{x}, \mathbf{y}; K) = \sum_{c=1}^{K} \sum_{i=1}^{n} \ln p_{\boldsymbol{\theta}^c}(y_i|\boldsymbol{x}_i, c)p_{\boldsymbol{\beta}}(c|\boldsymbol{x}_i).$$

The EM algorithm from Dempster et al. (1977) is an iterative method that can be used to estimate the *maximum a posteriori* (*MAP*) of $p(c|\boldsymbol{x}, y)$, since $c$ is latent, and alternates between the expectation (E-)step, which computes the relative membership of the observations into each segment, and a maximisation (M-)step, which computes the maximum likelihood estimate (MLE) of the parameters of the log-likelihoods of each segment's logistic regression and the tree structure. These new logistic regression and tree estimates are then used to determine the distribution of the latent variables in the next E-step. Considering the number of segments $K$ fixed, the E and M-steps of the EM can be derived as follows.

**E-step -** At iteration $(s+1)$, the partial membership of an observation $i$ to segment $c$ is:

$$p(c_i|\boldsymbol{x_i}, y_i; \boldsymbol{\theta}^{c(s)}, \boldsymbol{\beta}^{(s)}) = t_{i,c}^{(s+1)} = \frac{p_{\boldsymbol{\theta}^{c(s)}}(y_i|\boldsymbol{x_i})p_{\boldsymbol{\beta}^{(s)}}(c|\boldsymbol{x_i})}{\sum_{c'=1}^{K} p_{\boldsymbol{\theta}^{c'(s)}}(y_i|\boldsymbol{x_i})p_{\boldsymbol{\beta}^{(s)}}(c'|\boldsymbol{x_i})}.$$

For notational convenience, we denote the matrix of partial membership of all observations to all segments as $\mathbf{t} = (t_{i,c})_{1 \leq i \leq n, 1 \leq c \leq K}$.

**M1-step -** The previous E-step allows to derive the new MLE of the logistic regression parameters of each segment $c$ as:

$$\boldsymbol{\theta}^{c(s+1)} = \arg\max_{\theta^c} \mathbb{E}[\ell(\boldsymbol{\beta}, (\boldsymbol{\theta}^{c'})_1^K; \mathbf{x}, \mathbf{y}; K, \mathbf{t}^{(s+1)})|(\boldsymbol{\theta}^{c(s)})_1^K, \boldsymbol{\beta}^{(s)}, K]$$

$$= \arg\max_{\boldsymbol{\theta}} \sum_{i=1}^{n} t_{i,c}^{(s+1)} \ln p_{\boldsymbol{\theta}^c}(y_i|\boldsymbol{x_i}).$$

**M2-step -** Similarly, a new tree structure can be derived by the new MLE of its parameter $\beta$:

$$\boldsymbol{\beta}^{(s+1)} = \arg\max_{\boldsymbol{\beta}} \mathbb{E}[\ell(\boldsymbol{\beta}, (\boldsymbol{\theta}^c)_1^K; \mathbf{x}, \mathbf{y}; K, \mathbf{t}^{(s+1)})|\boldsymbol{\theta}^{c(s)}, \boldsymbol{\beta}^{(s)}, K]$$

$$= \arg\max_{\boldsymbol{\beta}} \sum_{i=1}^{n} \sum_{c=1}^{K} t_{i,c}^{(s+1)} \ln p_{\boldsymbol{\beta}}(c|\boldsymbol{x_i})$$

where $p_{\boldsymbol{\beta}}(c|\boldsymbol{x_i})$ is estimated by relative frequency in each leaf, such that $p_{\boldsymbol{\beta}}(c|\boldsymbol{x}) = \frac{|\mathbf{c}^{\mathcal{L}(\boldsymbol{x})}|}{|\boldsymbol{x}^{\mathcal{L}(\boldsymbol{x})}|}$, where $\mathcal{L}(\boldsymbol{x})$ denotes the leaf in which $\boldsymbol{x}$ falls. In this M2-step, one could argue that $\boldsymbol{\theta}^{c(s+1)}$ could be used, since it is computed in the M1-step, which could improve convergence. However, this would require recalculating the partial memberships $\mathbf{t}^{(s+1)}$. Hence it is unclear if this would be beneficial to the algorithm's runtime.

Additionally, tree induction methods like CART or C4.5 do not follow a maximum likelihood approach, so that they rather try to minimise a so-called impurity measure, the Gini index or the entropy, respectively. However, since it is hoped that segments $c^\star$ are "peaks" of the distribution $p_{\boldsymbol{\beta}}(c|\boldsymbol{x})$, we assume the log-likelihood can be approximated by the entropy:

$$\boldsymbol{\beta}^{(s+1)} \approx \arg\max_{\boldsymbol{\beta}} \sum_{i=1}^{n} \sum_{c=1}^{K} t_{i,c}^{(s+1)} \underbrace{p_{\boldsymbol{\beta}}(c|\boldsymbol{x_i})}_{\begin{cases} \approx 1 \text{ for } c = c^\star, \\ 0 \text{ otherwise.} \end{cases}} \ln p_{\boldsymbol{\beta}}(c|\boldsymbol{x_i}).$$

This last formulation allows to obtain $\boldsymbol{\beta}^{(s)}$ from a simple application of the C4.5 algorithm, with observations properly weighted by $t_{i,c}$. However, this approach suffers from two main drawbacks: first, all observations are used in all logistic regression $p_{\boldsymbol{\theta}^c}$ which might hinder runtime; second, all possible values of $K$ must be iterated through since the EM algorithm does not allow for the disappearance of a segment $c$ contrary to the SEM approach developed hereafter.

## 4.2 An SEM estimation strategy

Using an MCMC approach, a straightforward way of building logistic regression trees is to propose a tree structure, fit logistic regressions at its leaves, and evaluate the goodness-of-fit using Equation 3 of the resulting logistic regression tree. This is somehow the way LMT works: a tree structure is proposed based on C4.5, logistic regressions are fitted using the LogitBoost algorithm, and the tree is pruned back using a goodness-of-fit criterion. Doing so for all possible tree structures being intractable, we design a way of generating "good" candidates by relying on an SEM algorithm, which we call [MODEL]. The E-step of the previous section is thus replaced by a Stochastic (S-) step which has some consequences on the M-steps.

**S-step -** The "soft" assignment of the EM algorithm of the previous section is hereby replaced by a "hard" stochastic assignment such that:

$$c_i^{(s+1)} \sim p_{\boldsymbol{\theta}^{\cdot(s)}}(y_i|\boldsymbol{x_i})p_{\boldsymbol{\beta}^{(s)}}(\cdot|\boldsymbol{x_i}).$$

**M1-step -** Thanks to the previous step, the segments are now assigned such that the logistic regressions can be estimated using only observations affected to their segment:

$$\boldsymbol{\theta}^{c(s+1)} = \arg\max_{\boldsymbol{\theta}^c} \ell(\boldsymbol{\theta}; \mathbf{x}^{c(s+1)}, \mathbf{y}^{c(s+1)})$$

$$= \arg\max_{\boldsymbol{\theta}^c} \sum_{i=1}^{n} \mathbb{1}_c(c_i^{(s+1)}) \ln p_{\theta^c}(y_i|\boldsymbol{x}_i; c).$$

**M2-step -** This is again approximated by C4.5's (unweighted) impurity measure, the entropy, using only observations affected to each segment.

## 4.3 GOING BACK TO "HARD" SEGMENTS

### 4.3.1 *MAP* ESTIMATE

In the previous sections, we relaxed the discrete problem into "soft" assignments $p_{\boldsymbol{\beta}}(c_j|\cdot)$. This allows observations to "partly" belong to each segment, which can be interpreted as a mixture of logistic regressions: all observations are scored by all models which are subsequently weighted. This is arguably not interpretable, nor the initial goal to retrieve a tree such as in Figure 1. An assignment of each sample $i$ to a single most appropriate model, *i.e.* to a leaf of the segmentation tree, is achieved in parallel from the (S)EM algorithm(s) by a *MAP* step such that:

$$\hat{c}_i^{(s)} = \arg\max_c p_{\boldsymbol{\beta}^{(s)}}(c|\boldsymbol{x}_i).$$

### 4.3.2 LEAVES AS SEGMENTS

Alternatively, we can simply consider the leaves of the estimated tree $p_{\boldsymbol{\beta}^{(s)}}(c|\boldsymbol{x}_i)$ as segments. In other words, if we number the terminal nodes of the tree (*e.g.* left to right), $\hat{c}_i^{(s)}$ becomes the number of the leaf where $\boldsymbol{x}_i$ lands. There is no obvious reason why this would work better than the MAP estimation, nor a theoretical justification. However, experiments on simulated data in Section 5 suggest it performs better.

## 4.4 CHOOSING THE BEST SEGMENTATION CANDIDATE

The EM and SEM strategies introduced in the two previous sections for segmentation are merely "segments providers". Indeed, through the iterations 1 to $S$, as argued in the two preceding paragraphs, segmentations $\hat{\mathbf{c}}^{(1)}, \ldots, \hat{\mathbf{c}}^{(S)}$ are proposed through a *MAP* or *leaves as segments* rule parallel to these algorithms. The best performing segmentation $s^\star$ is then chosen using Equation 3 (where the search space is restricted to the proposed segmentations).

## 4.5 CONVERGENCE PROPERTY: EXPLORING THE NUMBER OF SEGMENTS

In the preceding sections, the number of segments $K$ was assumed to be fixed. However, the *MAP* scheme introduced in this section allows us, when going from "soft" $p_{\boldsymbol{\beta}}(c_j|\cdot)$ to "hard" segment assignment, to explore a number of segments potentially way lower than $K$: for a fixed segment $c$, if there is no observation $i$ such that $p_{\boldsymbol{\beta}}(c|\boldsymbol{x}_i) > p_{\boldsymbol{\beta}}(c'|\boldsymbol{x}_i)$ for $c' \neq c$, than the segment is empty, which is equivalent to producing a segmentation in $K-1$ segments. Supplemental to this thresholding effect, the use of an SEM algorithm makes it possible to enforce this phenomenon: as $c$ is drawn in the S-step, there is a non-zero probability of not drawing a particular segment $c$ at a given step $(s)$. When run long enough, the chain will stop with $K = 1$. This can be seen as a strength since it does not require to loop over the number of segments $K$ which would be required for an EM algorithm, which is why focus is given on the SEM algorithm in what follows.

For the *leaves as segments* approach, the number of segments $K$ entirely depends on the form (in particular the depth) of the tree. Little can be said about the quality of the exploration.

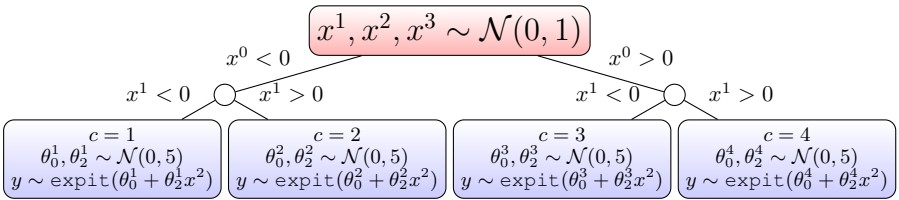

Figure 2: Data simulation procedure.

### 4.6 CATEGORICAL VARIABLES, DISCRETIZATION OF CONTINUOUS VARIABLES ON EACH SEGMENT AND VARIABLE SELECTION

As logistic regression assumes linearity of the log-odd ratio w.r.t. continuous features and conversely might estimate a coefficient relative to a categorical level taken by few samples with a lot of variance, practitioners often discretize continuous features and regroup categorical levels to obtain the best model. In parallel, the goal of the SEM algorithm is to split the population into segments that "behave" differently. Thus, to achieve better performance, both the discretization of continuous features and the grouping of levels of categorical features must be segment dependent. The variables selected in each segment by the logistic regression (*via* an L1-regularisation) will also be different. We therefore add a "processing" step to our SEM algorithm.

**P-step - Discretization of continuous features** We discretize continuous features using a Minimum Description Length Principle method (MDLP, see Fayyad & Irani (1993)), which consists in trying different cutting values (midpoints of distinct consecutive values), selecting the best based on entropy, and deciding whether it is worth continuing to further discretize with the MDLPC criterion.

**Merging levels of categorical features** We merge levels of categorical features using a $\chi^2$ method, where we compute the $\chi^2$ contingency of every unique pair of categories, merge the pair with the highest contingency if the contingency is above a certain value, and repeat (while keeping a minimum of two categories). In the following section, we generate continuous features and show empirical consistence of the proposed method.

## 5 PERFORMANCE ON SIMULATED DATA

### 5.1 DATA SIMULATION MECHANISM

We generate the data from a logistic regression tree (see Section 1.3) drawing features $X^j$, $j \in \{1, 2, 3\}$, from $\mathcal{N}(0, 1)$, forming a decision tree by choosing splits $x^0 = 0$ and $x^1 = 0$, which yields $K = 4$ segments, and drawing distinct logistic regression coefficients $\boldsymbol{\theta}^c$ from $\mathcal{N}(0, 5)$ on each of the leaves of the tree. Default $y$ is then drawn from $p_{\boldsymbol{\theta}^c}(\cdot|\boldsymbol{x}; c)$, see Figure 2.

### 5.2 IMPORTANCE OF THE HYPER-PARAMETERS & EMPIRICAL CONVERGENCE

The SEM algorithm, which pseudo-code can be found in Appendix C, has multiple parameters: using *MAP* or *leaves as segments*, the number of iterations $S$, the initial number of segments $K$, and, being an MCMC method, the number of initialisations. Indeed, to avoid risking a bad performance because of an unlucky initialisation, we randomly initialize the algorithm multiple times in parallel, run it and return the best model found among the parallel runs.

Figure 3 displays the results: its first row represents Equation 3 w.r.t. the number of initialisations, the number of iterations and the number of samples (which were resp. fixed at 5, 100 and 6,000 when another parameter was tested). As the BIC criterion is an information criterion, it is computed on the training set. There is no test set (as it would not make sense to penalize the likelihood on a test set). The second row displays the percentage of experiments where the correct tree (its depth, the features chosen to split and the splits themselves within the $[-0.1, 0.1]$ range). In both cases, the option to use a *MAP* estimation or *leaves* to obtain segments are plotted against each other. To obtain a 1-standard deviation, *i.e.* 68 %, confidence interval, each experiment is run 200 times.

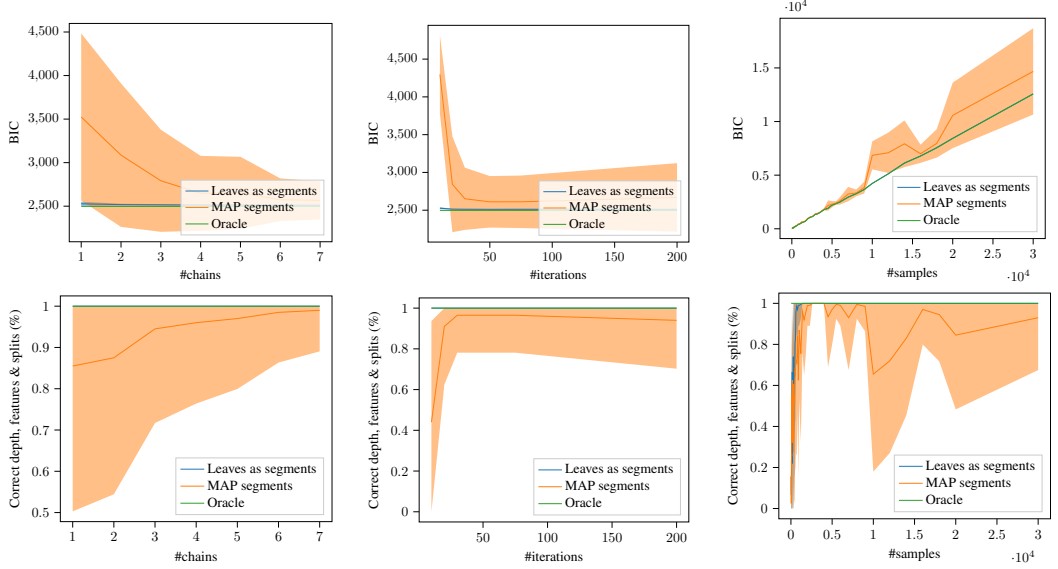

Figure 3: Top row: achieved sum of BIC (Equation 3); Bottom row: proportion of correct trees and models retrieved.

Table 1: Comparison of the proposed approach [MODEL]-SEM and other classical algorithms on three open-source datasets and our closed-source dataset.

| AUC | Logistic regression | Decision Tree | [MODEL] SEM | Gradient Boosting | Random Forest |
|---|---|---|---|---|---|
| Statlog (german) | 62.1 (3.6) | 56.6 (4.4) | 68.0 (2.4) | 63.3 (2.8) | 62.7 (3.1) |
| Adult | 84.1 (0.7) | 81.7 (0.9) | 85.3 (0.3) | 86.9 (0.2) | 84.8 (0.3) |
| Fraud | 96.9 (0.6) | 93.6 (0.4) | 95.9 (0.9) | 74.3 (1.8) | 97.3 (0.2) |
| In-house ($\pm$ vs current method) | -3.02 | -2.66 | -1.78 | -0.17 | +0.36 |

When increasing the size of the data set, we asymptotically get the model which we used to simulate the data more frequently and with greater confidence. Thus, this empirical convergence and consistency to the data generating mechanism allow us to be confident that the approach is correct on well-separated data. We will now apply this algorithm to real data.

## 6 PERFORMANCE ON REAL DATA

### 6.1 BENCHMARK ON OPEN-SOURCE DATASETS

As our in-house data, used in the following section, cannot be openly shared, we resort to some experiments on open-source datasets. The *statlog (german)* and *adult* datasets from UCI (Dua & Graff (2017)) are used as they both have mixed-type data, few features (20 and 14 resp.) and many observations (w.r.t. the number of features - 1,000 and 48,842 resp.), as well as the *Credit Card Fraud Detection* from Kaggle (Le Borgne et al. (2022) - 29 features and 284,807 observations).

### 6.2 COMPARISON TO THE CURRENT METHOD AND CLASSIC ALGORITHMS

**In-house data** We now use a representative sample of $n = 100,000$ [COMPANY] clients (proportionally taken from the population, *i.e.* from every existing segment), for which we know the repayment performance, as a training sample. We use $n = 14,600,000$ as a test sample. The data is initially preprocessed (see Section 4.6) for the whole population and not for each segment, as the computational cost increases dramatically.

Table 2: Comparison of the proposed approach [MODEL]-SEM and other logistic regression trees algorithms.

|  | SEM | LMT | MOB |
|---|---|---|---|
| # segment (current: 9) | 2 | 11 | 1 |
| AUC ($\pm$ vs current method) | -1.52 | -7.70 | -5.21 |

**Benchmark models**   As benchmark models, we rely on one hand on "weak" but explainable learners which form the basis of our approach: decision trees and logistic regression. On the other hand, we choose also gradient boosting and random forests which traditionally perform quite well on tabular data, in particular on credit risk data, but are not explainable enough to be used in production on credit risk use cases as of now.

**Experimental setup**   Five to twenty (depending on the dataset's size) 70/30 training/test splits are drawn so as to give an idea of the variance of the approach. AUC (and its standard deviation in parentheses) are given in Table 1. All the algorithms are compared in Table 1 in absolute terms of Area Under the ROC Curve (AUC), a common metric in credit risk, for open-source datasets, and in relative terms (to the current method), for the in-house dataset. The segment-dependent processing step described in Section 4.6 is not performed.

**Hyperparameter tuning**   A grid-search is performed for our model as well as gradient boosting and random forest. For our model, for computational reasons, we split the training dataset further into a training and a validation set. The best set of hyperparameters is then chosen w.r.t. the highest AUC on the validation set. For standalone logistic regression, gradient boosting and random forest, a cross-validation grid-search is performed. Pseudo-code as well as the hyperparameter grids are available in Appendix B.

**Results**   First, as expected, [MODEL]-SEM performs better than both the logistic regression or the decision tree alone. Other methods such as Gradient Boosting or Random Forest have similar or better results than the current method, but when comparing the results on each existing segment we see that they don't perform well on all segments universally. Because of that, even with a slightly better overall performance, we don't consider them a more satisfying method.

Comparing [MODEL]-SEM to the current method, we can see that its performance is satisfactory. However, the current method has a "wider" hypothesis space, since the data preprocessing (discretization, grouping categories for categorical variables) is done separately for every segment and involved a lot of manual fine-tuning. Additionally, our "small" sample of 100k observations (w.r.t. the test sample of 14.6M observations) cannot capture "very small", manually-crafted segments (several segments with approx. 50-100k observations in the full sample, *e.g.* voluntary associations/unions). The current method thus achieves higher performance.

Our SEM algorithm nevertheless creates fewer segments than the current method (the result depends on the initialisation, but we usually find 2 to 4 segments, compared to the 9 segments of the current method). Thus, the proposed model is less complex, requires much less of practitioners' time, but yields lower performance.

**Comparison to existing logistic regression trees algorithms**   We compare the SEM algorithm to the other logistic regression tree approaches discussed in Section 2, except for LOTUS which does not have any available implementation. We now use the version of SEM which incorporates the data preprocessing in each segment (see Section 4.6). For 10 segments, 100 iterations and 5 different initialisations, this amounts to discretizing and merging levels 5000 times. Adding this segment-dependent data preprocessing does slightly increase our performance at the cost if increased computation time. The small number of segments is limiting the performance, but is nevertheless superior to existing approaches (see Table 2).

## 7 CONCLUSION

This paper aims at formalising an old problem in *Credit Scoring*, namely client segmentation, by providing a literature review as well as a new algorithmic approach. As is often the case, practitioners have had good intuitions to deal with practical and theoretical requirements, such as performing clustering techniques, choosing segments empirically from the resulting visualisation and fitting logistic regression on these. However, situations can easily be imagined where such practices can fail, which is why other methods, which take into account the predictive task, shall be preferred. To this end, a new method is proposed, based on an SEM algorithm.

On simulated data, it shows good results which demonstrate empirical consistency of the approach. On open-source real data, it shows superior performance than "plain" logistic regression and decision tree. On real data from [COMPANY], the automatic [MODEL]-SEM almost competes with the current performance which requires a lot of human time and expert knowledge, and performs better than the other existing logistic regression trees approaches. The [MODEL] method is available as a package from [MASK], as well as scripts to reproduce results from Sections 5, 6.1.

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

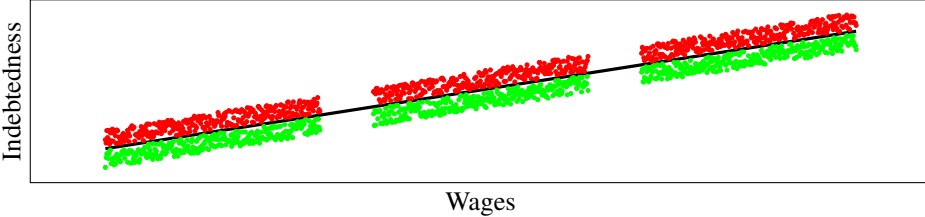

(a) Multi-modal wages and indebtedness data generating mechanism with $y = \{0, 1\}$ classes displayed in red and green respectively.

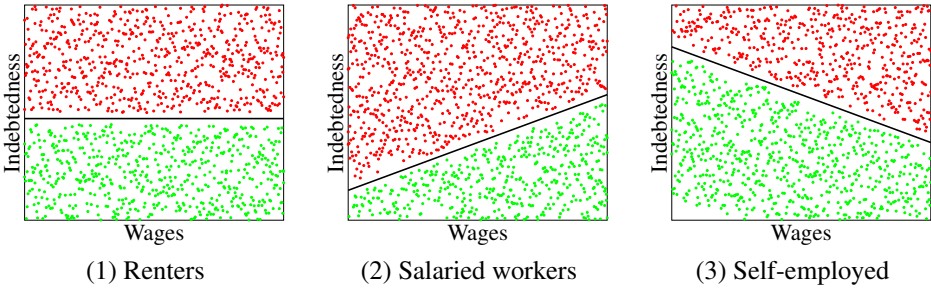

| (1) Renters | (2) Salaried workers | (3) Self-employed |

(b) Uni-modal wages and indebtedness data generating mechanism with $y = \{0, 1\}$ classes displayed in green and red respectively w.r.t. a "source of wages" categorical feature.

Figure 4: Ad hoc practices failing situations.

Marc Sumner, Eibe Frank, and Mark Hall. Speeding up logistic model tree induction. In *European Conference on Principles of Data Mining and Knowledge Discovery*, pp. 675–683. Springer, 2005. doi: 10.1007/11564126_72.

Achim Zeileis, Torsten Hothorn, and Kurt Hornik. Model-based recursive partitioning. *Journal of Computational and Graphical Statistics*, 17(2):492–514, 2008. doi: 10.1198/106186008X319331.

## A  AD-HOC PRACTICES CAN FAIL

It appears clearly that this approach does not directly optimise a predictive goal. The segmentation and the logistic regression are optimised independently, according to different error losses. When those goals don't align, theses practises can fail.

The first of these failing situations is when the probability density function of covariates (suppose for simplicity that all of them are continuous) $p(\boldsymbol{x})$ is multi-modal as on Figure 4a where we distinguish the lower, middle and upper-classes of respective low, average and high wages and indebtedness. An unsupervised generative approach like PCA would urge the practitioner to construct 3 models (one for each of the aforementioned classes). However, displaying $y = 1$ (default) as red and $y = 0$ as green, we can see that perfect separation can be achieved: it depends solely on the indebtedness level (the ratio of wages over indebtedness). Thus, the resulting models would be asymptotically the same. Since each of the 3 models would have three times less samples to learn from, it would amount to increasing the estimation variance of the coefficients, and ultimately result in lower performance. On a practical note, one could argue that it reduces interpretability by adding an avoidable complexity to the decision system.

The second failing situation is the counterpart of the first tailored data generating mechanism and is displayed in Figure 4b. This time, suppose the covariates are uniformly sampled. Suppose there is a third categorical feature "wages source" which is drawn from three levels: renters, salaried workers and self-employed. Suppose that renters' risk level do not depend on their indebtedness, which is typically low, salaried workers' risk level is positively correlated with their indebtedness ratio as was the case for the first introductory example and self-employed people's risk level is negatively correlated with this indebtedness ratio. In this situation, an unsupervised generative "clustering"

algorithm like the projection of the data on the two first PCA axes would not partition the data and we would construct only one scorecard. This scorecard would have high model bias since it is too simple to accommodate for the variety of the data generating mechanism.

## B HYPERPARAMETER SEARCH

**Standalone Logistic Regression**   An Elastic-net penalty is applied. The best set of hyperparameters, consisting in the regularization strength and the balance between L1 and L2 regularizations is found *via* 5-fold cross-validation with scikit-learn.

**[MODEL]-SEM Logistic Regression for** $c$   An L2-penalty is applied with a (small) regularization parameter of $0.01$. The rationale behind this choice is to avoid overfitting.

**[MODEL]-SEM Logistic Regression for** $\hat{c}$   An L1-penalty is applied with a (small) regularization parameter of $0.01$. The rationale behind this choice is that some features might not be relevant for some segments.

**Gradient Boosting**   Grid-search is performed with scikit-learn *via* 5-fold Cross-Validation with the following hyper-parameters:

- `learning_rate`: [0.01, 0.1, 0.5, 2.0],

- `n_estimators`: [100, 300, 1000],

- `subsample`: [0.5, 0.75, 1.0],

- `min_samples_split`: [10, 30],

- `min_samples_leaf`: [1, 5, 20],

- `max_depth`: [2, 5, 10],

- `max_features`: ['log2', 'sqrt', None]

**Random Forest**   Grid-search is performed with scikit-learn *via* 5-fold Cross-Validation with the following hyper-parameters:

- `n_estimators`: [10, 100, 1000],

- `min_samples_split`: [10, 30],

- `min_samples_leaf`: [1, 5, 20],

- `max_depth`: [2, 5, 10],

- `max_features`: ['log2', 'sqrt', None]

- `ccp_alpha`: [0.0, 0.1, 1.0]

## C [MODEL]-SEM: PSEUDO-CODE

**Data:** Features, targets, number of segments, number of iterations: $\mathbf{x}$, $\mathbf{y}$, K, max_iter
**Result:** Final number of segments, assignments of each instance to a segment, logistic
      regression parameters of each segment, segmentation tree: $K^{\star}, \mathbf{c}^{\star}, \{\boldsymbol{\theta}^{c^{\star}}\}_{1 \leq c^{\star} \leq K^{\star}}, \beta^{\star}$
Initialize "soft" ($\mathbf{c}^{(0)}$) and "hard" ($\hat{\mathbf{c}}^{(0)}$) segments at random: $c_i, \hat{c}_i \sim [1, \ldots, K]$;
Track best iteration so far with best_iter, best_BIC $= 0, \infty$;
**while** $s \leq$ *max_iter* **do**
    **for** $1 \leq c \leq K$ **do**
        $(\mathbf{x}^{c(s)}, \mathbf{y}^{c(s)})$ is the dataset of instances for which $c_i = c$;
        **(M1)** Estimate $\boldsymbol{\theta}^{c(s)}$ by fitting a logistic regression to $(\mathbf{x}^{c(s)}, \mathbf{y}^{c(s)})$;
        Estimate $\hat{\boldsymbol{\theta}}^{\hat{c}(s)}$ by fitting a logistic regression to $(\mathbf{x}^{\hat{c}(s)}, \mathbf{y}^{\hat{c}(s)})$;
    **end**
    **(M2)** Estimate $\boldsymbol{\beta}^{(s)}$ by fitting a decision tree to $\mathbf{x}, \mathbf{c}^{(s)}$;
    **(S)** Draw new $\mathbf{c}^{(s+1)}$ from a row-wise normalization of $p_{\boldsymbol{\theta} \cdot}(\mathbf{y}|\mathbf{x}) p_{\boldsymbol{\beta}^{(s)}}(\cdot|\mathbf{x})$
    (*note*: some segments might not be drawn at all, thus $K \leftarrow$ distinct values in $\mathbf{c}^{(s+1)}$);
    Calculate "hard" segments $\hat{\mathbf{c}}^{(s+1)}$ as the row-wise MAP of $p_{\boldsymbol{\theta} \cdot}(\mathbf{y}|\mathbf{x}) p_{\boldsymbol{\beta}^{(s)}}(\cdot|\mathbf{x})$;
    If sum of BICs (Equation 3) $\leq$ best_BIC, update best_iter, best_BIC to current values;
**end**
Output $K^{\star}, \mathbf{c}^{\star}, \{\boldsymbol{\theta}^{c^{\star}}\}_{1 \leq c^{\star} \leq K^{\star}}, \beta^{\star}$ which correspond to the lowest sum of BICs (as given by
  best_BIC).

**Algorithm 1:** [MODEL]-SEM pseudo-code.

