# OpenReview forum: "Tree-structure segmentation for logistic regression"
_ICLR.cc/2023/Conference — Submitted to ICLR 2023_

### Official Review · Reviewer_tW1b · 2022-10-21

**Confidence:** 3
**Correctness:** 2
**Technical Novelty And Significance:** 2
**Empirical Novelty And Significance:** Not applicable
**Recommendation:** 3

**Clarity, Quality, Novelty And Reproducibility:**

quality: it is not clear to me whether the derivation of the method proposed in the paper is rigorous. I also find the experimental evaluation somewhat limited and unconvincing.

clarity: I have concerns about the clarity of the paper. The authors need to provide more details about the method derivation. Notation needs to be improved as well to cover all the concepts they introduce. Their writing can be improved too. For example, the author can consider providing an overview of section 4 at the beginning of the section. Some key terms appeared not to be explained. For example, what does SEM stand for?

originality: the proposed method seems to be new to my knowledge. Nonetheless, taking a EM approach to deal with this problem is somewhat non-surprising either.

**Strength And Weaknesses:**

Pros:
1. the paper is concerned with an important problem in the financial industry.
2. the proposed method is interesting, where the joint learning of both the segment and the model of each segment seems to be a more principled approach compared to the more prevalent naive two-stage practice.
3. The paper provides a review of several existing methods for this problem.

Cons:
1. Many mathematical derivations seem to be omitted. I cannot find a supplemental file with a more detailed derivation. For example, in p(x,y) on page 3, why the second equality is true? In the first equation of section 4.1, it is not clear to me how this equation can be derived from equation 4.1. The derivation of the EM algorithm is also not clear, despite the EM algorithm being the key part of the paper. I would suggest the authors to provide as many derivation details as possible to better justify the correctness of their approach. As it stands, it is difficult to evaluate with high certainty that the mathematics of the paper is correct.

2.  the notation of the paper can be improved. For example, I cannot find what beta stands for. The definition of BIC is also not specified.

3. the experimental results are some what limited and unconvincing. For example, experiments are only run on four real world datasets. It is not clear whether the proposed method has better performance or not compared to alternatives. Even though the authors argue that the proposed method provides more consistency across segments. It is not clear to me whether this is true across all datasets or is true only about the in-house dataset.

**Summary Of The Paper:**

The paper deals with the credit scoring problem. The paper formulates the problem as a logistic regression tree problem, where the data points are split into segments and each segment has a logistic regression model to model the probability of default. The paper then proposes an algorithm that can jointly learn segment assignments and the logistic regression model of each segment in an expectation maximization fashion. Experiments are conducted on both synthetic data and real-world data to evaluate the performance of the proposed method.

**Summary Of The Review:**

Overall, while I think the paper deals with an important and interesting real-world problem, I believe the presentation of the paper can be improved and experiments can be more exhaustive to reach a publishable stage. Therefore, I recommend rejection.

---

> ### Author Response · Authors · 2022-11-18
> **Comments on review tW1b**
>
> Thank you for your careful review of our paper and interesting comments. Following the order of your cons:
>
> 1. - W.r.t. the omitted mathematical derivations, we have added two steps to Equation 1 (now Eq. 2) which should justify your concern with p. 3. In Section 4.1 the maximum likelihood is given by how we modeled the data in what is now Eq. 1 (sorry for the prior absence of / mislabelling). The derivation of the EM algorithm follows from the fact that the partial membership $t_{i,c}$ is the conditional distribution of c given observations $x_i$, $y_i$ and current values of the parameters $\theta$ and $\beta$ (which would be a simple proportion in the case of Gaussian mixtures).
> 2. - W.r.t. notations, $\beta$ and BIC, indeed the beta notation can be inexplicit for a decision tree. It is there to emphasize that it is a different model than the logistic regressions parametrized by $theta^c$, fitted separately. We consider the "parameters" of the decision tree to be its split features and cut-points. A sentence has been added to Section 3. The BIC (Bayesian Information Criterion similar to AIC) has also been introduced in Section 3.
> 3. - W.r.t. the experiments being "limited", we understand that the current trend is to provide benchmarks on lots of open datasets. However, fitting the 5 models on a moderately small dataset (e.g. "adult") is very computationally intensive (roughly 1 day) since we perform, for 5 to 20 train / test shuffles (to obtain confidence intervals), a "standard" train / validation grid-search for the hyperparameters of our model and cross-validation grid-search for Gradient Boosting and Random Forests. Subsequently, within the time constraint of Stage 1 Discussion and our available computing resources, it was unrealistic to plan more experiments.
>
> On other topics:
> - "For example, the author can consider providing an overview of section 4 at the beginning of the section." This overview has been added to the revised paper.
> - "For example, what does SEM stand for?": as explained in the abstract, "we will conduct estimation using a Stochastic-Expectation-Maximisation (SEM) algorithm"; thus SEM is an EM algorithm where a single segment / class c is drawn from the distribution $t_{i,c}^{(s+1)}$. Our addition to Section 4 should also help clarify that.
>
> Again, sincere thanks for your helpful comments.

---

### Official Review · Reviewer_hAgH · 2022-10-24

**Confidence:** 3
**Correctness:** 3
**Technical Novelty And Significance:** 2
**Empirical Novelty And Significance:** 2
**Recommendation:** 3

**Clarity, Quality, Novelty And Reproducibility:**

It would help to clarify the implementation of baselines used as well as hyperparameter tuning used.

Why does the model name need to be redacted?

**Strength And Weaknesses:**

This paper presents an interesting approach for regression with the following strengths and weaknesses:

**Strengths**
* The paper presents an interesting methodological approach, it is a clear an elegant model, which has a rich and challenging inference task.
* The proposed approach appears to be quite effective, especially on the corporate dataset.

**Weaknesses**
* I feel that there is a wide landscape of related approaches related to mixture of experts, especially tree-structured mixture of experts [1,2, inter alia] that is not explored. There is also much work on tree structured methods that do not split on specific features e.g. [3,4,5 inter alia] I think that it would be very helpful to consider a wider landscape of related method.
* I wonder if the authors could clarify the selection of baseline algorithms? Would it make sense to try a simple few layer MLP? Would it make sense to use xgboost? Kernel ridge regression?
* Empirical analysis - The results are most impressive on the corporate dataset, it would be helpful to understand why the proposed approach does not work as well and what characteristics of data would lead someone to select the proposed approach compared to the other methods.
* The proposed approach is a nice model, but it leaves many open questions. For instance, What are its theoretical properties? When should someone prefer it to other models? How does the quality of the structure effect performance? When is the structure learnable? I think without a deeper understanding of the proposed approach. It is not clear to me whether we can justify the modeling approach on its own without such understanding or more complete empirical analysis.

[1] Bishop, Christopher M., and Markus Svensén. "Bayesian hierarchical mixtures of experts." UAI, 2003.

[2] Zhao, Wenbo, Yang Gao, Shahan Ali Memon, Bhiksha Raj, and Rita Singh. "Hierarchical routing mixture of experts." In 2020 25th International Conference on Pattern Recognition (ICPR), 2021.

[3] Choromanska, Anna E., and John Langford. "Logarithmic time online multiclass prediction." Advances in neural information processing systems 28 (2015).

[4] Daumé III, Hal, Nikos Karampatziakis, John Langford, and Paul Mineiro. "Logarithmic time one-against-some." In International Conference on Machine Learning, pp. 923-932. PMLR, 2017.

[5] Yu, Hsiang-Fu, Kai Zhong, Jiong Zhang, Wei-Cheng Chang, and Inderjit S. Dhillon. "PECOS: Prediction for enormous and correlated output spaces." Journal of Machine Learning Research 23, no. 98 (2022): 1-32.

**Summary Of The Paper:**

This paper presents a tree structured model for regression for applications in finance. The authors describe an inference procedure for fitting the structure and parameters. The authors provide empirical analysis of proposed method and its variants on their own corporate data and academic datasets.

**Summary Of The Review:**

While the proposed approach is an interesting and elegant model, there are concerns about where it fits into a broader landscape of related work. Further, better understanding when and why the proposed approach is effective would greatly strengthen the paper.

---

> ### Author Response · Authors · 2022-11-18
> **Comments on review hAgH**
>
> Thank you for your careful review of our paper and interesting comments. Following the order of your remarks:
>
> - W.r.t. the previous works on Hierarchical Mixtures of Experts (HME), which is a common remark among all reviewers, we feel that our approach and HME only have a "tree-like" structure in common but differ in many (if not all) other aspects: similar to our (discarded) EM approach (see Section 4.1), all models perform prediction, with different weights, which is arguably not as intepretable as a logistic regression tree (hence our Section 4.3 to obtain ``hard'' segments); additionally, splits are multivariate which further hinders interpretability, contrary to a split feature associated to a cut point in classical decision trees. All in all, such a logistic regression tree can be considered as a constrained version of HME. A similar remark has been added to the revised paper in Section 2.
> - W.r.t. the choice of baseline algorithms, we rephrased a sentence in the original paper under the new ``Benchmark model'' paragraph. We chose to use two relatively weak learners which form the basis of logistic regression trees: logistic regression and decision tree; hence logistic regression trees should perform at least as good as both methods. In credit risk, interpretability is key, thus more complex methods are very rarely used. Nevertheless, we wished to see if logistic regression trees could compete with other, less interpretable / more flexible tree-based models such as gradient boosting and random forests. Neural networks could have been considered but often lag behind on tabular data and we didn't consider it necessary to have many "useless" benchmarks (in so far as they will never be used in practice due to their complexity).
> - W.r.t. your open questions, and in particular the context in which to use this method, we could simply answer (although not satisfactory, to say the least) ``whenever the use of such simple models is mandatory'' such as in credit risk. Thus, it is not a necessity from the theoretical / performance viewpoint, but strictly from an interpretability / regulatory one.
>
> Again, sincere thanks for your helpful comments.

---

### Official Review · Reviewer_heWb · 2022-10-26

**Confidence:** 3
**Correctness:** 1
**Technical Novelty And Significance:** 2
**Empirical Novelty And Significance:** 2
**Recommendation:** 3

**Clarity, Quality, Novelty And Reproducibility:**

The paper is quite clear, but pseudo-code for the final SEM algorithm, including the locally applied pre-processing steps, would help greatly.

**Strength And Weaknesses:**


The empirical comparison is unsatisfactory. LMT and MOB, the most direct competitors, are only compared to on the private data, not the public datasets.

The number of datasets included in the experiments is unnecessarily small.

In the evaluation of the proposed method, discretization and a method for merging categorical values are applied. It is not stated whether these methods are applied in conjunction with the other learning algorithms included in the experiments. It seems important, in order to make the comparison as fair as possible, to evaluate whether these pre-processing methods also help the other learning algorithms (or are detrimental to their performance if they are applied).

The proposed method appears to be computationally quite expensive, but this is not quantified in the paper. In particular, it seems the maximum number of leaves will have to be quite limited to apply the proposed algorithm in practical applications.

It seems important to relate the proposed method to the classic work on mixtures of experts.

Hyperparameter settings need to be clearly stated. For example, the poor results for gradient boosting are a strong indication that it was inappropriately configured. Also, it appears that L_1 regularization may have been used for logistic regression in the proposed method (see Section 4.6). How was the strength of regularization determined? And was similar regularization applied in stand-alone logistic regression?


**Summary Of The Paper:**

The submission presents an algorithm for learning decision trees with logistic regression models at the leaf nodes. It initially considers soft splits and estimation of the model using expectation maximization before proceeding to more interpretable hard splits and a stochastic expectation maximization algorithm for learning the trees. An important parameter of the algorithm is the maximum number of leaf nodes in the resulting tree. AUROC is estimated for three UCI datasets and one private credit-rating dataset. On the private dataset, the proposed method outperforms LMT and MOB, which both also grow decision trees with logistic regression models.

**Summary Of The Review:**

The proposed method may have merit, but the submission does not present sufficient empirical evidence to be convincing. There is no discussion of related work on mixtures of experts. Limitations of the proposed method, particularly considering runtime, are not discussed.

---

> ### Author Response · Authors · 2022-11-18
> **Comments on review heWb**
>
> Thank you for your careful review of our paper and interesting comments. Following the order of your remarks:
>
> - We acknowledge that LMT and MOB should also be compared to the public datasets but we failed to do so within the time constraint of submission.
> - W.r.t. the number of datasets being "unnecessarily small", we understand that the current trend is to provide benchmarks on lots of open datasets. However, fitting the 5 models on a moderately small dataset (e.g. "adult") is very computationally intensive (roughly 1 day) since we perform, for 5 to 20 train / test shuffles (to obtain confidence intervals), a "standard" train / validation grid-search for the hyperparameters of our model and cross-validation grid-search for Gradient Boosting and Random Forests. Subsequently, within the time constraint of Stage 1 Discussion and our available computing resources, it was unrealistic to plan more experiments.
> - W.r.t. the preprocessing steps being applied or not to the benchmark approaches, ideally we would do both: data is either discretized / merged beforehand and all the models are fitted, or data is not pre-processed and the preprocessing eventually takes place at each segment for our approach, or not at all for the benchmark models. However, for computational reasons and as discussed in the original paper in Section 6, the results in Table 1 are obtained with a global preprocessing, while results of Table 2 are obtained with a segment-dependent preprocessing (hence slightly better results for [MODEL]-SEM in Table 2 w.r.t. the in-house dataset compared to Table 1), but no uncertainty could be quantified.
> - W.r.t. the computional cost of our method, we acknowledge that the absence of its quantification is detrimental to our paper and plan on reporting, in the (likely) event of a resubmission, the training times (average, standard) of all models of Table 1. Qualitatively, if pre-processing is not performed at a segment-level (see previous paragraph), the training time with a reasonable number of segments (say 5) is in the same order of magnitude than gradient boosting.
> - W.r.t. the previous works on Hierarchical Mixtures of Experts (HME), which is a common remark among all reviewers, we feel that our approach and HME only have a "tree-like" structure in common but differ in many (if not all) other aspects: similar to our (discarded) EM approach (see Section 4.1), all models perform prediction, with different weights, which is arguably not as intepretable as a logistic regression tree (hence our Section 4.3 to obtain ``hard'' segments); additionally, splits are multivariate which further hinders interpretability, contrary to a split feature associated to a cut point in classical decision trees. All in all, such a logistic regression tree can be considered as a constrained version of HME. A similar remark has been added to the revised paper in Section 2.
> - W.r.t. hyperparameter tuning, and more broadly the setup of the experiments, we have added Appendix B to the revised paper.
>
> Again, sincere thanks for your helpful comments.

---

> > ### Comment · Reviewer_heWb · 2022-11-24
> > **Acknowledgement of Comments on review heWb**
> >
> > Thank you for the responses to my comments. I am glad to see that they were helpful.

---

### Decision · Program_Chairs · 2023-01-20

**Decision:**

Reject

**Justification For Why Not Higher Score:**

No originality of interest to other researchers.

**Justification For Why Not Lower Score:**

May be useful for the authors' company.

**Metareview: Summary, Strengths And Weaknesses:**

(a) The paper uses expectation-maximization (EM) to train a tree with a small number of leaves, with a logistic regression model at each leaf.

(b) Could be useful in practice, for interpretability.

(c) No algorithmic originality. Trees with different models at their leaves, and EM to learn a mixture model, are well-known. Moreover the paper is naive from a business perspective: we need to predict loss given default (LGD) in addition to probability of default. We should also predict the probability of the loan being restructured, and the economic cost to the bank of that, even when there is no formal default.

The citations are to old literature from 2014 and before except for a handbook, so this paper is not part of the ICLR research domain. The authors should also learn from relevant academic business research, such as https://arxiv.org/abs/1906.01174.



**Summary Of Ac-Reviewer Meeting:**

No meeting.